# Robust salient object detection based on triple attention-guided multi-resolution fusion and feature refinement

Geng Wei[ID][1]*, Mi Zhou[1], Jian Sun[1], Xiao Shi[1], Ming Yin[2], Xinran Zhao[1], Xueyao Lin[1]

1 School of Physics and Electronics, Nanning Normal University, Nanning, China, 2 School of Electronic Science and Engineering, South China Normal University, Foshan, China

* wei_geng@nnnu.edu.cn

## Abstract

Salient object detection (SOD) is dedicated to highlighting the critical image elements in complex visual scenes. However, this task faces two serious challenges: first, salient objects are often submerged in cluttered backgrounds and are susceptible to disturbance from background noise; second, the substantial scale variation among these objects presents considerable challenges for accurate detection. In order to address these challenges, an attention-based method for salient object detection is proposed. Firstly, we innovatively design a Triple Attention-guided Multi-resolution Fusion (TAMF) module, which integrates a spatial, channel, and global attention mechanism to dynamically adjust feature weights to suppress background noise. At the same time, it introduces a multi-resolution feature fusion framework to enhance cross-scale interactions. Secondly, we propose the Feature Refinement (FR) module, which utilizes four parallel convolutional branches and different-scale dilated convolutions in conjunction with the triple attention mechanism to precisely detect and enhance the salient object features, as well as effectively address the challenges of scale changes. Evaluations across five challenging benchmark datasets demonstrate notable improvements over advanced methods, highlighting our model's effectiveness and competitive advantage. Code is available at: https://github.com/zbbany/ATMF_FRNet.git.

## Introduction

Salient Object Detection (SOD) focuses on identifying regions within an image that capture human visual attention. SOD is an efficient preprocessing method that has important applications in various fields, including image retrieval [1], visual tracking [2], video segmentation [3], and intelligent image cropping and editing [4]. Recent advancements driven by Convolutional Neural Networks (CNNs) have significantly propelled SOD research, empowering models to learn both abstract semantic concepts and fine-grained details. Predominant CNN frameworks typically employ a

**Data availability statement:** All underlying data (the minimal dataset) and code required to replicate the findings of this study are publicly available from the Zenodo repository (DOI: 10.5281/zenodo.18138097).

**Funding:** This research was funded by the Natural Science Foundation of Guangxi Province (Grant No. 2020GXNSFAA297184) and the National Natural Science Foundation of China (Grant No. 62161031). All expenses incurred in the preparation of this manuscript were covered by the funders, which are public welfare-oriented scientific and technological management departments. The funders had no role in the study design, data collection and analysis, decision to publish, or preparation of the manuscript.

**Competing interests:** The authors have declared that no competing interests exist.

pyramidal design. In this structure, features from higher layers encapsulate rich semantic content, albeit at the cost of losing some spatial detail, while features from lower layers retain higher spatial fidelity and capture localized information more precisely. Typically, high-level features facilitate the coarse identification of salient regions, whereas low-level features, rich in spatial cues, are better suited for refining object boundaries. Therefore, researchers have proposed a variety of novel network architectures to achieve multi-level feature fusion. Among them, U-shaped structures have attracted the most attention since they have the ability to generate comprehensive feature representations by incorporating top-down pathways into standard bottom-up classification backbones.

One common limitation of the U-shaped structure is that when the global semantic information collected at the highest level is gradually transmitted along a top-down path, it may be disrupted and gradually weakened by a large number of local patterns at lower levels. To address this problem, [5–7] proposed aggregating semantic information extracted from high-level features into feature maps at each feature level. However, indiscriminately aggregating high-level information across all levels can complicate the learning process and potentially impair the subsequent decoding. Furthermore, due to the changes in different scales of salient objects, the SOD model requires high sensitivity to changes in salient objects. In response to these situations, various improved methods have been proposed, including optimizing the feature map using a cyclic mechanism [8], introducing additional constraints in the saliency map [9], and integrating multi-scale feature information [9,10].

Building upon this foundation, our work delves into the application of attention mechanisms within U-shaped networks. On the one hand, we adopt an attention-guided noise suppression strategy to achieve cross-layer fusion of high-level features, instead of directly aggregating salient location information. This alleviates the dilution problem of high-level semantic information without introducing noise from high-level features to other feature layers. On the other hand, the low-level feature maps preserve abundant localized details essential for precise visual analysis. To fully leverage this information to enhance the sensitivity of our method to scale variations of salient objects, we introduce a mechanism for multi-scale feature enhancement to maximize the potential of low-level feature maps. The key innovations presented in this article can be briefly described below:

1. A novel Triple Attention-guided Multi-resolution Fusion (TAMF) module is introduced. It suppresses background noise by adjusting the spatial and channel weights of global features. Meanwhile, the model's adaptability to multi-scale objects is enhanced by parallel multi-resolution feature cross-fusion.
2. A Feature Refinement (FR) module is designed. It employs multi-scale parallel convolutional branches alongside the triple attention mechanism to dynamically boost the model's sensitivity towards scale variations and noise disturbances, thereby effectively restoring intricate feature details.
3. Comprehensive evaluations and ablation studies on five widely-used SOD benchmarks substantiate the effectiveness and competitive edge of our introduced approach.

## Related works

### Salient Object Detection

Salient Object Detection (SOD) is a fundamental task in computer vision, aiming to highlight visually salient regions in images. According to input characteristics and modalities, it can be divided into SOD for natural images (including RGB, RGB-D, and RGB-T) and SOD for Optical Remote Sensing Imagery (ORSI).

In natural image SOD, RGB-based methods are the most basic, relying on color information to distinguish salient regions. Their performance is limited in scenarios with low contrast and cluttered backgrounds due to the lack of spatial complementary information. Significant progress has been made in deep learning-based RGB SOD research: Zhang et al. [11] explored the recursive construction of saliency maps using multi-scale features to enrich target representation; Feng et al. [12] proposed an attentional feedback loop mechanism to accurately capture structural details of salient objects; Kong et al. [13] designed a feature propagation technique to integrate multi-scale information and enhance the model's robustness in complex environments; and the authors of [14] proposed an Integrity Cognitive Network to improve the integrity of salient objects in prediction results. RGB-D SOD introduces depth information to provide spatial cues, becoming a research hotspot. Its core progress focuses on the optimization of cross-modal fusion and multi-scale learning: Sun et al. [15] proposed CATNet, which hierarchically fuses RGB and depth features through a cascaded and aggregated Transformer; Hu et al. [16] designed a cross-modal fusion and progressive decoding network to refine fused features for accurate localization; Zhong et al. [17] proposed MAGNet, which integrates multi-scale awareness and global fusion mechanisms to adapt to objects of different scales; LESOD [18] is a lightweight RGB-D network that balances detection accuracy and computational efficiency; Gao et al. [19] proposed an adaptive fusion and attention regulation method to dynamically adjust the fusion weights of features and suppress background interference. RGB-T SOD is a branch of multi-modal SOD for natural images, fusing RGB and thermal infrared modalities. Thermal infrared images capture the temperature characteristics of objects and are insensitive to changes in illumination and occlusion, making them suitable for harsh scenarios such as low light. The core challenge lies in fusing complementary information and suppressing cross-modal noise.

ORSI SOD faces unique challenges due to the top-down viewing angle and complex ground objects of remote sensing images, making natural image SOD methods inapplicable. In recent years, specialized methods have emerged: ORSID-iff [20] accurately models salient regions in remote sensing images based on diffusion models, achieving excellent performance on small and scattered targets; Han et al. [21] explored a lightweight and efficient network, optimizing feature extraction and parameter configuration to achieve efficient and accurate detection in ORSI.

Despite the progress in various SOD branches, limitations remain in complex scenarios such as strong background interference and large target scale variations. This paper proposes an attention-based RGB SOD method to address these issues, providing an accurate and robust detection solution.

### Attention mechanism

Attention mechanisms are fundamental to human perception, especially within the visual system. Recently, researchers in this field have extensively explored and successfully applied the attention mechanism, which has achieved outstanding results in various tasks such as SOD, sequence learning, pedestrian re-identification, and image recovery. To elaborate, in the domain of SOD, attention mechanisms have been effectively utilized in the design of networks for robust RGB-T segmentation [22]. Beyond this specific task, the principle of attention guidance also plays a pivotal role in broader areas such as few-shot learning, where it aids in discriminative feature learning [23], meta-regularization [24], multi-view encoding for action recognition [25], and facilitates knowledge transfer between large multimodal models [26]. In image classification tasks, researchers have made equally remarkable progress. For instance, Qin et al. [27] introduced the 'squeeze and excite' (SE) operation, skillfully modeling inter-channel relationships to boost model efficacy. In addition to channel-wise attention, CBAM [28] similarly introduced spatial attention. This dual mechanism addresses channel dependencies

while also evaluating the significance of spatial information, enabling the model to concentrate on critical image areas by adaptively modulating feature weights across spatial positions. Such self-attention mechanisms, applied across both channel and spatial axes, effectively accentuate informative feature regions and salient channels, consequently elevating model recognition precision.

### High-level feature learning

Despite the significant development of SOD technology, the current mainstream methods still mainly focus on integrating and enhancing low-level features to accurately represent object boundaries. However, this strategy unintentionally ignores the importance of high-level feature learning, leading to a relative lack of exploration of high-level features in the field of SOD. To compensate for this deficiency, numerous techniques have adapted established semantic segmentation tools, such as the Pyramid Scene Parsing (PSP) [29] and its derivative versions, to enhance the model's understanding of high-level features. However, it should be pointed out that there are important differences between SOD and semantic segmentation tasks. This difference makes it difficult to directly apply semantic segmentation modules to SOD tasks to achieve the required salient object localization accuracy, and they can only achieve suboptimal performance.

## Methods

### Overview

Fig 1 illustrates the structure of the framework introduced in this paper. ResNet50 [31] serves as the backbone, extracting multi-scale initial features S1, S2, S3, S4. The resolutions of the initial features decrease gradually, and their downsampling factors relative to the input image are 4, 8, 16, and 32, respectively. We then introduce the Triple Attention-guided

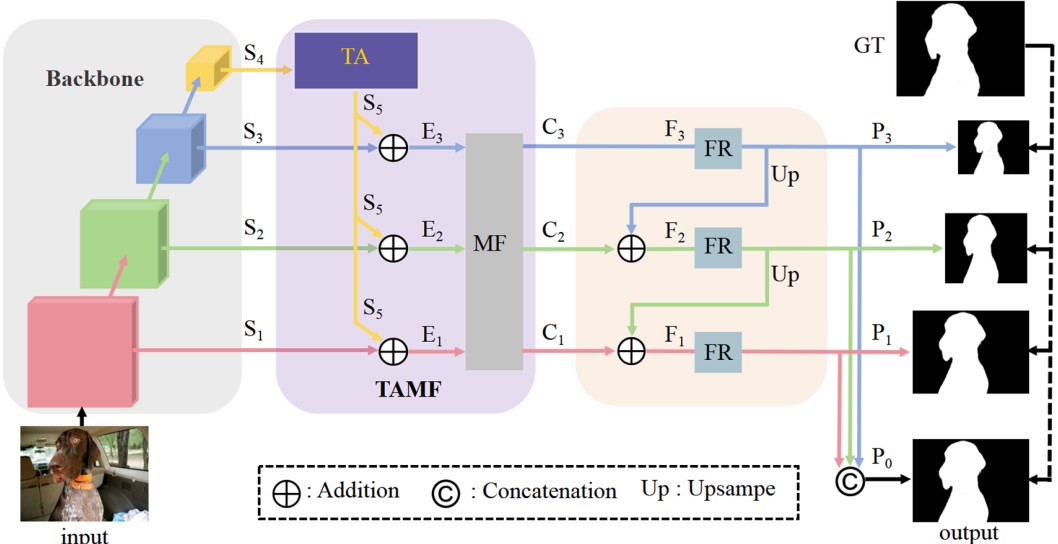

**Fig 1. The proposed model framework.** It has three main components: the backbone, the triple attention-guided multi-resolution fusion (TAMF) module, and the feature refinement (FR) module. The TAMF is used for the coarse localization of salient objects and for improving the fusion effect and complementarity of cross-scale features, and FR is for recovering object details. *(The input images and ground truth (GT) annotations are sourced from the DUTS dataset (official download link: http://saliencydetection.net/duts/; source code repository: https://github.com/scott89/WSS, [30]). The dataset is licensed under the 3-Clause BSD Open Source License (compatible with the CC BY 4.0 license). The feature maps are generated by the code independently developed in this paper, and the model architecture diagram is created using Microsoft PowerPoint. All the above elements are original content by the authors and are licensed under the CC BY 4.0 license.)*

Multi-resolution Fusion (TAMF) module, which comprises two sub-modules: Triple Attention (TA) enhancement and Multi-resolution Fusion (MF), collectively enhancing the representation power of salient features. The TA sub-module effectively suppresses background noise by dynamically adjusting the spatial and channel weights of global features. At the same time, we incorporate the location information of noise suppression into each feature layer to initially localize the salient object area. The MF sub-module exchanges information across multi-resolution representations to obtain a preliminary salient feature map. Finally, a Feature Refinement (FR) module refines the preliminary salient feature map by combining multi-scale convolution and attention mechanisms, focusing on salient objects at different scales.

## Triple Attention-guided Multi-resolution Fusion (TAMF) module

**Triple Attention (TA) enhancement sub-module.** When conducting in-depth research on the issue of position information dilution caused by the layer-by-layer transmission of high-level features in U-shaped architecture, we found that the classic methods [5–7] introduce noise from high-level features when aggregating them into feature maps at different layers, which increases the difficulty of feature learning. To address this problem, we carefully designed the TA sub-module. This sub-module employs the triple attention mechanism, i.e., channel, spatial, and global attention, to perform adaptive enhancement and noise suppression of features. Specifically, the TA sub-module utilizes spatial attention to pinpoint key spatial locations, channel attention to filter important channel features, and global attention to enhance semantic information. After spatial and channel features are extracted in parallel, they are deeply fused with the global context features via element multiplication to establish robust feature constraints, which effectively suppress the background noise in the high-level features. Consequently, this approach avoids the problem of feature aggregation noise interference inherent in existing methods.

As shown in Fig 2(a), the input feature for TA is $S_4$. We compute the channel attention (CA) and spatial attention (SA) in parallel to generate channel weights and spatial weights, i.e., $S_{ca}$ and $S_{sa}$ respectively, for the input feature $S_4$. These weights are then applied to adjust the weights of each spatial location and channel in the input feature map, thereby enhancing key spatial locations and important channels, while correspondingly suppressing secondary locations and non key channels. This process achieves adaptive adjustment of the channel and spatial importance of the input feature map by dynamically adjusting the channel and spatial weights. Eventually, the obtained spatial and channel enhancement

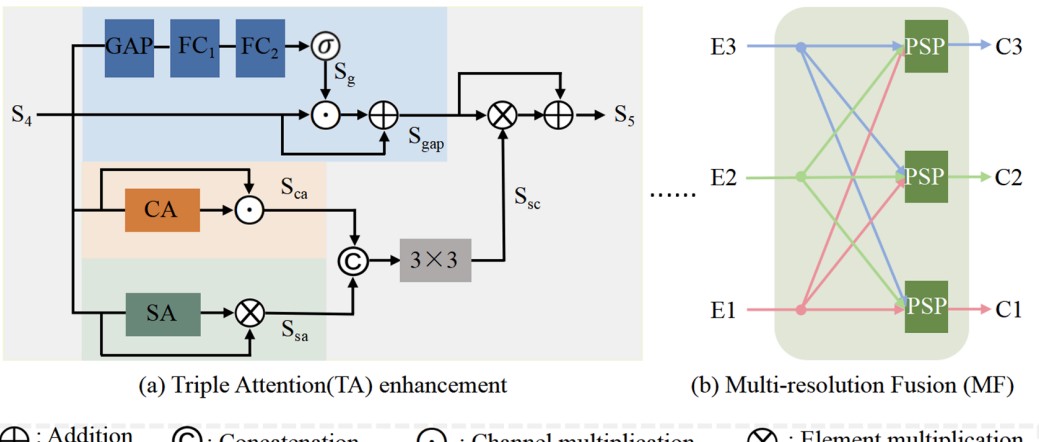

(a) Triple Attention(TA) enhancement    (b) Multi-resolution Fusion (MF)

$\oplus$ : Addition    $\copyright$ : Concatenation    $\odot$ : Channel multiplication    $\otimes$ : Element multiplication

**Fig 2. Structure of two key sub-modules TA and MF.** GAP, SA, and CA refer to global average pooling, spatial attention mechanism, and channel attention mechanism, respectively. All of them include convolution, BatchNorm, and ReLU.

features are fused to obtain $S_{sc}$. The process can be formulated as follows.

$$S_{ca} = CA(S_4) \odot S_4, \tag{1}$$

$$S_{sa} = SA(S_4) \otimes S_4, \tag{2}$$

$$S_{sc} = Conv_{3\times3}(Concat[S_{ca}, S_{sa}]), \tag{3}$$

where CA and SA denote channel attention and spatial attention processes, respectively, $\odot$ and $\otimes$ represent channel multiplication and element multiplication, respectively. We further downsample $S_4$ into a feature vector using Global Average Pooling (GAP) to obtain global image features.

$$S_g = \sigma(FC_2(ReLU(FC_1(GAP(S_4))))), \tag{4}$$

where $\sigma$ denotes the activation function of the sigmoid, $FC_1$ and $FC_2$ denotes the fully connected layers and $ReLU$ is an activation function. The value range $S_4$ is mapped to [0,1] using the sigmoid function. Although the $S_g$ vector can summarize the overall content of the input image, it cannot be directly used to reconstruct the original image since the spatial details of the image are lost, i.e., each eigenvalue is no longer associated with a specific position in the image. In other words, although this eigenvector contains global information about the image, its spatial details have been lost, making the decoding process impractical. Instead, we employ $S_g$ as a gate for channel-wise re-weighting to recalibrate the features and refine them via a residual connection.

$$S_{gap} = S_g \odot S_4 + S_4, \tag{5}$$

We use the obtained spatial channel fusion feature $S_{sc}$ as a gate to modulate the spatial channel attention in the global feature, which is regarded as a kind of constraint mechanism. This process can effectively suppress the background noise in the global feature and achieve feature refinement through the residual connection. Finally, the optimized feature $S_5$ is aggregated into the feature map of all feature levels. The process can be formally represented as:

$$S_5 = S_{gap} \otimes S_{sc} + S_{gap}, \tag{6}$$

$$E_i = Up(S_5) + S_i, i = \{1, 2, 3\}, \tag{7}$$

Overall, we effectively learn and capture a general overview of the entire image by introducing a spatial, channel, and global triple attention mechanism that effectively suppresses background noise and significantly highlights salient features in the image.

**Multi-resolution Fusion (MF) sub-module.** Current methods mostly adopt a bottom-up feature integration strategy, which hinders the complete utilization of distinctive characteristics across feature layers at different scales, leading to limited capability in representing complex salient objects. Different from existing methods, this paper employs adaptive multiresolution feature fusion, which significantly strengthens the model's capacity for perceiving multi-scale salient features by promoting the interaction and complementarity among cross-scale feature layers. Subsequently, we deeply refine these fused features using the PSP [29] network to obtain a preliminary salient feature map. With its powerful context capture

 

capability, the PSP network greatly enriches the semantic representation of feature maps and provides strong support for subsequent accurate segmentation. As shown in Fig 2(b), the inputs to MF are $E_1$, $E_2$, $E_3$. The process is outlined below.

$$C_k = PSP\left(\sum_{i=1}^{3} F(E_i, E_k)\right), k = 1, 2, 3, \tag{8}$$

where $F(E_i, E_k)$ denotes the up-sampling or down-sampling operation from resolution $E_i$ to $E_k$. Specifically, the up-sample operation refers to a 1 × 1 convolution layer followed by a bilinear interpolation, while a 3 × 3 convolution layer with a stride of 2 refers to the down-sample operation. If $E_i = E_i$, $F(\cdot)$ denotes a 3 × 3 convolution layer without sampling layers. psp$(\cdot)$ denotes the PSP [29] network.

In summary, by introducing a multi-resolution fusion sub-module, our approach not only enhances the ability of each scale-specific layer to perceive multi-scale salient features but also significantly improves the accuracy and robustness of SOD.

### Feature Refinement (FR) module

In the task of SOD, the model must exhibit exceptional sensitivity to scale variations of salient objects, considering their substantial size differences. Existing studies [3] and [32] have shown that using dilated convolution with different scales can effectively extend the network's perceptual range, help the network capture features of objects with different sizes, and avoid excessive image distortion. Inspired by these research results, we designed four parallel convolutional branches in the proposed feature refinement module, each of which is unique.

Each branch consists of a standard k × k convolution kernel and a 3 × 3 dilated convolution with a dilation rate of d. The values of k and d are set to be 1, 3, 5, and 7, respectively, as shown in Fig 3. This design approach allows each branch to focus on capturing salient object features of specific shapes and scales. Thus, the FR module can comprehensively cover and respond to salient objects at different scales and significantly enhances the model's multi-scale feature

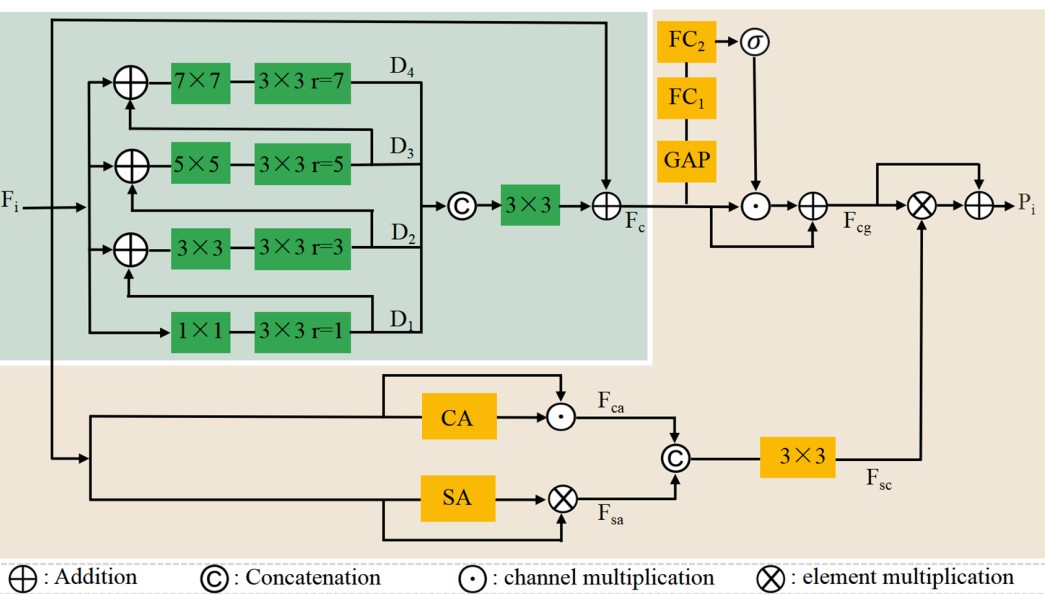

**Fig 3. Structure of feature refinement (FR) module.** The meanings of GAP, SA, and CA are the same as before.

perception capability. In order to further enhance the information flow and complementation between parallel branches, we innovatively introduce the inter-branch hopping mechanism. This mechanism boosts internal branch communication and ensures each learns distinct features, thereby enhancing feature extraction efficiency and efficacy. The output of each branch, where the convolutional kernel size is defined as $k_n = 2n - 1$, is described as follows:

$$D_n = \begin{cases} DConv_{3\times3, 2n-1}(Conv_{k_n \times k_n}(F_i)), n = 1 \\ DConv_{3\times3, 2n-1}(Conv_{k_n \times k_n}(F_i \oplus D_{n-1})), n = 2, 3, 4 \end{cases}, \tag{9}$$

where $D_n$ is the output of the nth branch, $Conv_{k_n \times k_n}$ is the standard convolution with kernel $k_n \times k_n$, $DConv_{3\times3, 2n-1}$ is the dilated convolution with a kernel size of 3 and a dilation rate 2n-1, and $\oplus$ represents addition. As can be seen from above, we use smaller convolution kernels to process the first branch to preserve more detailed features, while for the later branches, we gradually increase the size of the convolution kernels to better capture the overall regional features. This design method not only preserves the feature details of long-term dependencies but also ensures the integrity of the overall feature representation. Notably, by employing convolution kernels of varying scales across branches, the FR module can flexibly adapt to salient features at multiple scales, greatly improving the detection ability of the model and making it perform well in complex scenes. Subsequently, the four branches are concatenated, denoted as:

$$F_c = Conv_{3\times3}(Concat[D_1, D_2, D_3, D_{4,}]) \oplus F_i, \tag{10}$$

where $Concat$ denotes the concatenation operation and $Conv_{3\times3}$ denotes a 3×3 convolution.

In addition, we also incorporate the proposed triple attention mechanism to further improve the robustness of feature connections and reduce the impact of background noise. This process can be expressed as follows:

$$F_{ca} = CA(F_i) \odot F_i, \tag{11}$$

$$F_{sa} = SA(F_i) \otimes F_i, \tag{12}$$

$$F_{sc} = Conv_{3\times3}(Concat[F_{ca}, F_{sa}]), \tag{13}$$

$$F_{cg} = \sigma(FC_2(ReLU(FC_1(GAP(F_c))))) \odot F_c + F_c, \tag{14}$$

$$P_i = F_{cg} \otimes F_{sc} + F_{cg}, \tag{15}$$

where the meanings of CA, SA, $\odot$, $\otimes$, and $\sigma$ are the same as above, and the detailed formulation is presented in the corresponding section of the TAMF module. This attention mechanism enhances the focusing ability of feature maps on salient object regions and the expression ability of semantic information.

By processing different branches in parallel, the FR module can capture more complete multi-scale feature information. Introducing cross-layer connections and attention mechanisms significantly improves the effectiveness and quality of network architecture in feature extraction, along with the stability and adaptability of the model across diverse scenarios.

## Loss function

This article adopts a hybrid loss function, including BCE loss and IoU loss [33], that is, the overall loss of the proposed method i.e, $\mathcal{L}_{CPR}$ can be correspondingly expressed as, $\mathcal{L}_{CPR}(P, G) = \mathcal{L}_{BCE} + \mathcal{L}_{IoU}$ The $\mathcal{L}_{BCE}$ and $\mathcal{L}_{IoU}$ are computed from the ground truth $G$ and the prediction map $P$, and the specific formulas for them are given as follows

$$\mathcal{L}_{BCE} = -\sum_{x=1}^{H}\sum_{y=1}^{W} G(x, y)log(P(x, y)) + (1 - G(x, y))log(1 - P(x, y)), \tag{16}$$

$$\mathcal{L}_{IoU} = 1 - \frac{\sum_{x=1}^{H}\sum_{y=1}^{W} P(x, y)G(x, y)}{\sum_{x=1}^{H}\sum_{y=1}^{W} P(x, y) + G(x, y) - P(x, y)G(x, y)}, \tag{17}$$

where $W$ and $H$ are the width and the height of the image, respectively. In the model, $G(x,y)$ and $P(x,y)$ denote the ground truth label and the predicted saliency label in the current location $(x,y)$, respectively. The training phase employs a multilevel supervision mechanism. To guarantee consistency between the training process and inference output with the ground truth map, the channel count for each feature map is reduced to one. Concurrently, the spatial resolution of these feature maps is resized to align with the input image dimensions.

## Results

### Implementation details

All experiments are run on publicly available PyTorch 1.10.0 and Python 3.8 platforms, with training and testing performed on two Intel® Xeon® Gold 6230 CPUs, 64 GB of 2933 MHz RAM, and an RTX 4080 SUPER GPU. The input images are uniformly resized with a resolution of 352 × 352. During the model training, the data augmentation techniques, including normalization, random cropping, and mirroring, are applied. The encoder backbone is initialized with ResNet-50 [31] or PVTv2 [34] pre-trained weights (the rest of the model is initialized with reference to [35]). The chosen optimizer is SGD, configured with an initial learning rate of 0.05, a weight decay of 5e-4, an eps of 1e-8, and a momentum value of 0.9. To manage the learning rate schedule, we implement a combined warm-up and linear decay strategy. The training process trains for 53 epochs with a batch size of 32 (ResNet-50) or 10 (PVTv2). Gradient clipping is also applied to mitigate the risk of gradient explosion.

### Datasets and evaluation metrics

Model training is carried out on the DUTS-TR dataset [30], which includes 10,553 samples and serves as the standard SOD training benchmark. The model evaluation is performed on five commonly used benchmark datasets: ECSSD [36] (1,000 images with complex semantic contexts), PASCAL-S [35] (850 images with multiple salient objects derived from the semantic segmentation dataset), HKU-IS [37] (4,447 images with various objects in the foreground), OMRON [32] (5,168 images with complex structural objects), and DUTS-TE [30]. In this context, the DUTS dataset is divided into two subsets: the training set (DUTS-TR, 10,553 images) and the test set (DUTS-TE, 5,019 images), all of which are labeled with pixel-level labels.

This study provides a comprehensive comparative analysis of our model with the leading current models using five evaluation metrics:

1. S-measure ($S_m$) [38] primarily assesses the similarity of structures, similar to human visual perception. $S_m = ms_o + (1 - m)s_r$ i.e., it combines object-perceived similarity $s_o$ and region-perceived similarity $s_r$, weighted by m = 0.5 (m is a balancing coefficient).

2. E-measure ($E_\xi$) [39] is calculated as $E_\xi = \frac{1}{W \times H} \sum_{x=1}^{W} \sum_{y=1}^{H} \theta(\xi)$, where $\theta(\xi)$ reflects the alignment enhancement effect and $\xi$ is an alignment matrix that combines values of local pixels with the image-level average. In this study, the average E-measure ($E_\xi^m$) is used as the final measure to evaluate the model.

3. Weighted F-measure ($F_\beta^\omega$) [40] simplifies the $F_\beta$ metric with the formula $F_\beta^\omega = \frac{(1+\beta^2)Precision^\omega Recall^\omega}{\beta^2 Precision^\omega + Recall^\omega}$. $F_\beta^\omega$ is widely used in model evaluation, especially when dealing with complex scenes and small salient objects. $F_\beta^\omega$ is suitable for non-binary evaluations and assigns different weights ($\omega$) to errors by considering location and neighborhood information.

4. MAE ($M$) measures the average pixel-wise difference between the ground truth ($G$) map and the predicted saliency ($P$) map, both of which are first normalized to the range [0,1]. It is calculated as $M = \frac{1}{W \times H} \sum_{x=1}^{W} \sum_{y=1}^{H} |P(x,y) - G(x,y)|$.

5. FNR is used as an indicator for false negative assessment, and its calculation formula is $FNR = \frac{\sum_{x=1}^{W} \sum_{y=1}^{H} FN(x,y)}{\sum_{x=1}^{W} \sum_{y=1}^{H} G(x,y)} \times 100\%$, where $FN(x,y) = \begin{cases} 1, & \text{if } G(x,y) = 1 \ \& \ P(x,y) = 0, \\ 0, & \text{otherwise,} \end{cases}$, represents the number of positive examples that the model predicts incorrectly. As shown in Fig 6, FNR can effectively reflect the completeness of the forecasting results.

## Ablation study

To systematically assess the efficacy and contribution of individual components within our model, extensive ablation studies were performed on three benchmark datasets: ECSSD, PASCAL-S, and DUTS (results summarized in Table 1). All ablation variants maintained identical network architecture and hyperparameter settings.

We conducted five ablation experiments, i.e., baseline, baseline+TA, baseline+MF, baseline+TAMF, and baseline+FR, where "baseline" represents the benchmark for comparison, which uses ResNet50 [31] as the backbone and does not include the proposed module. Experimental findings demonstrate that incorporating the TA (Table 1, row 2) yields substantial gains over the baseline (row 1). TA effectively enhances the robustness of features by suppressing background noise and dynamically adjusting the spatial and channel weights of the global view. The visualization results in Fig 4 show that the results with TA (see column d) more effectively suppress background interference compared to those without TA (see column c).

The adoption of MF (row 3) also significantly improves the baseline performance. The performance of baseline + TAMF (row 4) outperforms the configurations where TA or MF are introduced alone, indicating that TA is well-compatible with MF and TAMF contributes significantly to the model. Integrating the FR module (row 5) also delivers considerable performance enhancement over the baseline. Fig 5 illustrates that for salient object detection, our method (see column g) consistently outperforms other methods over a wide scale range (the scale of objects from top to bottom increases in Fig 5). This performance advantage primarily stems from the FR module's four parallel convolutional branches, which effectively address the problem of large differences in salient object scales. Crucially, the combined baseline+TAMF+FR

**Table 1. Ablation study of the proposed model on ECSSD, PASCAL-S, and DUTS.**

| No | baseline | TAMF | | FR | ECSSD | | | | PASCAL-S | | | | DUTS | | | |
|---|---|---|---|---|---|---|---|---|---|---|---|---|---|---|---|---|
| | | TA | MF | | $S_m\uparrow$ | $E_\xi^m\uparrow$ | $F_\beta^\omega\uparrow$ | $M\downarrow$ | $S_m\uparrow$ | $E_\xi^m\uparrow$ | $F_\beta^\omega\uparrow$ | $M\downarrow$ | $S_m\uparrow$ | $E_\xi^m\uparrow$ | $F_\beta^\omega\uparrow$ | $M\downarrow$ |
| 1 | √ | | | | .902 | .923 | .849 | .051 | .828 | .855 | .739 | .089 | .825 | .845 | .692 | .073 |
| 2 | √ | √ | | | .924 | .951 | .909 | .034 | .852 | .890 | .803 | .070 | .880 | .913 | .815 | .042 |
| 3 | √ | | √ | | .920 | .946 | .902 | .037 | .853 | .894 | .803 | .067 | .877 | .913 | .815 | .042 |
| 4 | √ | √ | √ | | **.929** | .954 | .916 | .032 | .853 | .894 | .806 | .067 | .882 | .915 | .821 | .041 |
| 5 | √ | | | √ | .924 | .948 | .908 | .034 | .857 | .892 | .807 | .067 | .880 | .914 | .820 | .040 |
| 6 | √ | √ | √ | √ | **.929** | **.955** | **.919** | **.031** | **.865** | **.902** | **.824** | **.063** | **.891** | **.925** | **.841** | **.037** |

The best results are highlighted in bold. "↑" / "↓" means the higher/lower the indicator value, the better.

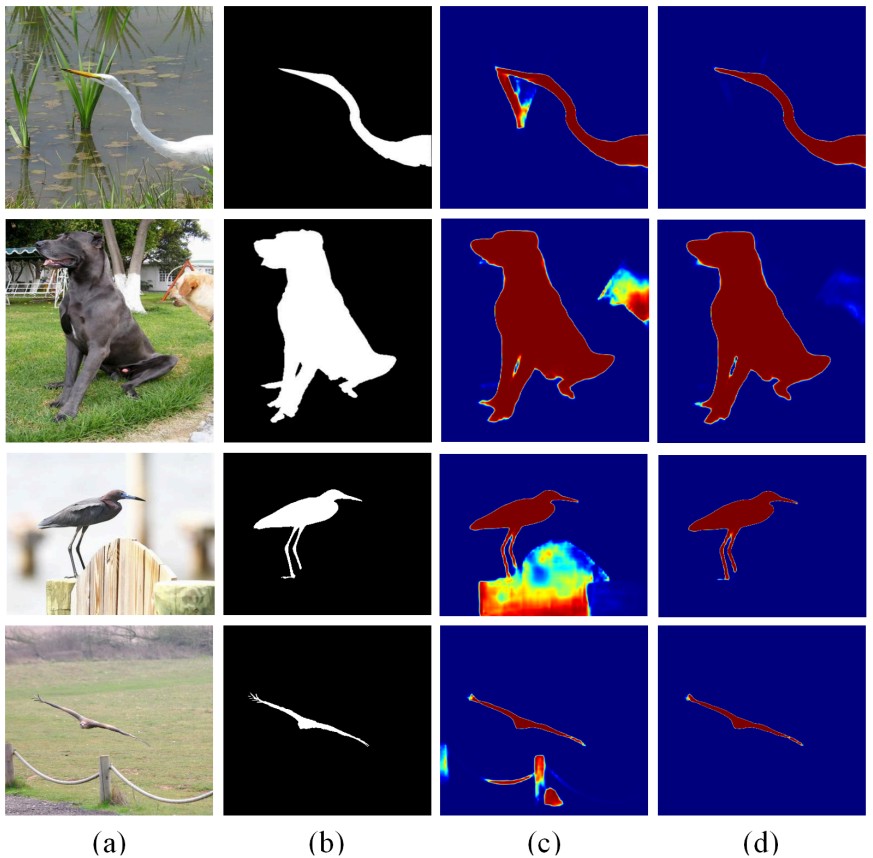

(a) (b) (c) (d)

**Fig 4**. **Visualization of the feature mapping with or without using TA.** (a) Input image. (b) Ground truth. (c) Without TA. (d) With TA. We can see that our TA method suppresses background noise better while highlighting objects. *(All feature maps are generated by the code in this paper. The input images and ground truth annotations are derived from the DUTS dataset (official download link: http://saliencydetection.net/duts/; source code repository: https://github.com/scott89/WSS, [30]). The dataset is licensed under the 3-Clause BSD Open Source License (compatible with the CC BY 4.0 license), and its licensing terms have been strictly followed in this work.)*

setup exceeds the performance of all other configurations listed in rows 2-5 of Table 1, confirming the high compatibility between TAMF and FR and validating the effectiveness of the proposed modules.

In addition to validating the overall effectiveness of the TAMF and FR modules in Table 1, we further conducted ablation studies on their internal key designs. As shown in Table 2, we replaced the proposed TA module with the classic dual-attention mechanism CBAM. The experimental results demonstrate that "+ TA + MF" significantly outperforms "+ CBAM + MF," confirming that the global attention branch and the specific fusion strategy in the TA module provide more effective feature enhancement compared to the standard channel-spatial attention mechanism (CBAM). Meanwhile, an ablation study on the inter-branch skip mechanism in the FR module is presented in Table 3. The results show that "+ FR (w/ Skip)" achieves statistically significant improvements across all metrics over "+ FR (w/o Skip)," which uses only standard parallel dilated convolution, thereby proving the effectiveness of cross-branch information interaction.

To verify the optimality of the kernel size/dilation rate (K/D) parameter selection in the FR module and analyze the model's sensitivity to variations in such parameters, we designed the ablation experiments presented in Table 4. Herein, N0 serves as the baseline scheme with the K/D set 1,3,5,7, and N1~N4 are variants of the baseline with scaled branches pruned or extended. By comparing the performance of these five distinct K/D scale sets, we validate the rationality of the

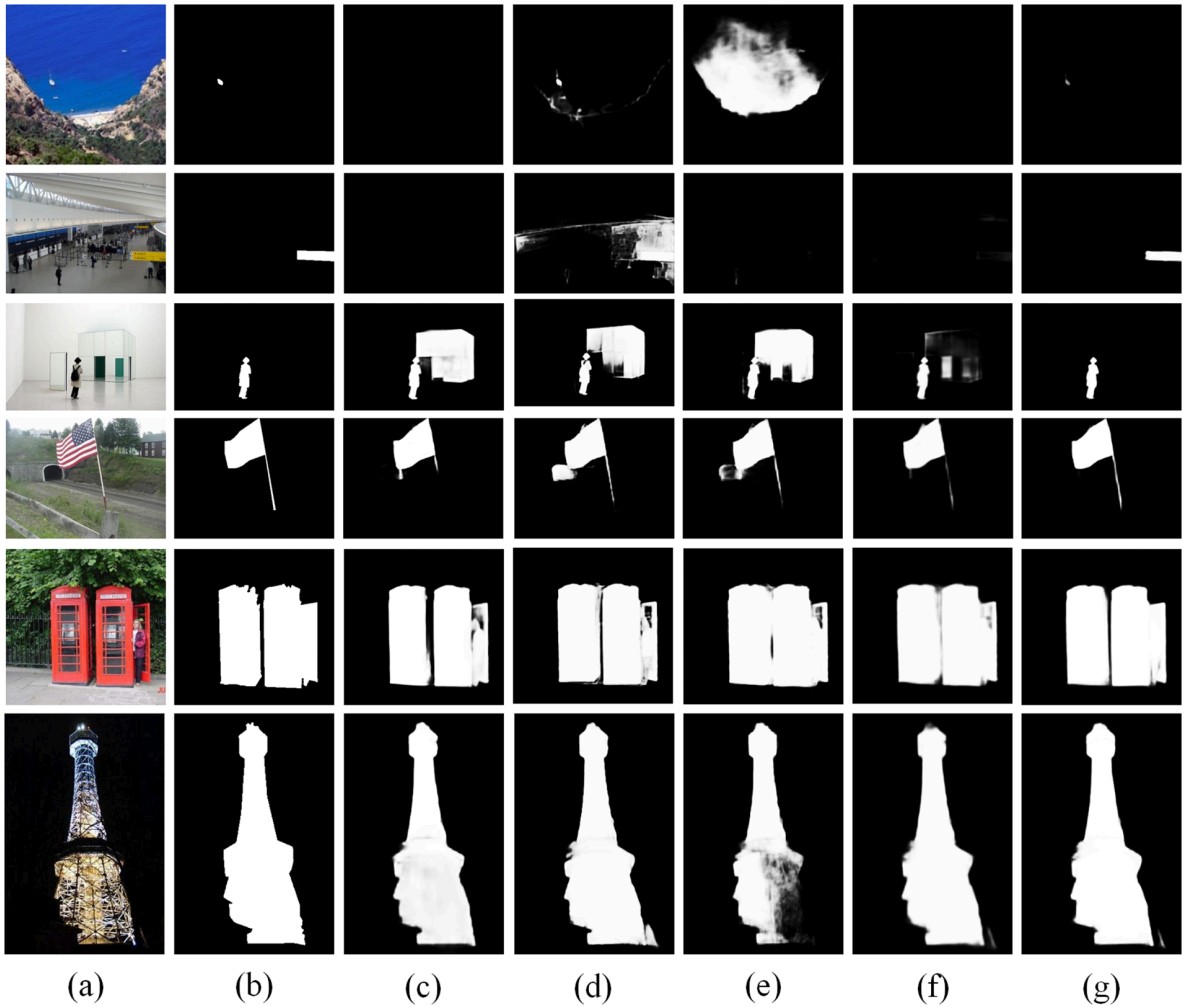

**Fig 5**. **Visual comparison results of our method with other methods at six different scales.** (a) Input image; (b) Ground truth; (c) F3Net; (d) ITSD; (e) MINet; (f) GCPANet; (g) Ours. As observed, our approach demonstrates a stronger capability in suppressing background noise and is more effective in addressing the challenges posed by highly variable scales and the sensitivity of SOD to such variations.

**Table 2**. **Ablation Study on TA vs. CBAM.**

| No | | ECSSD | | | | PASCAL-S | | | | DUTS | | | |
|---|---|---|---|---|---|---|---|---|---|---|---|---|---|
| | | $S_m \uparrow$ | $E_\xi^m \uparrow$ | $F_\beta^\omega \uparrow$ | $M \downarrow$ | $S_m \uparrow$ | $E_\xi^m \uparrow$ | $F_\beta^\omega \uparrow$ | $M \downarrow$ | $S_m \uparrow$ | $E_\xi^m \uparrow$ | $F_\beta^\omega \uparrow$ | $M \downarrow$ |
| 1 | baseline | .902 | .923 | .849 | .051 | .828 | .855 | .739 | .089 | .825 | .845 | .692 | .073 |
| 2 | +CBAM+MF | .922 | .946 | .906 | .036 | **.853** | .892 | .802 | .068 | .881 | .914 | .820 | **.041** |
| 3 | +TA (w/o Global) + MF | .927 | .952 | .914 | .033 | **.853** | .891 | .805 | .070 | .881 | .913 | .820 | .043 |
| 4 | +TA+MF | **.929** | **.954** | **.916** | **.032** | **.853** | **.894** | **.806** | **.067** | **.882** | **.915** | **.821** | **.041** |

The best results are highlighted in bold.

four-branch configuration 1,3,5,7 and elucidate the reasons for not adopting fewer/more branches or larger scales. Experimental results show that all performance metrics of N1~N3 are inferior to those of N0, indicating that the fine-grained scale 1, medium scale 5, and large scale 7 are critical scales required by the model, and the absence of any single scale will lead to feature loss: removing scale 1 results in the loss of texture details of small targets, skipping scale 5 causes

**Table 3.** Ablation Study on FR Module with/without Inter-Branch Skip Connections.

| No | | ECSSD | | | | PASCAL-S | | | | DUTS | | | |
|---|---|---|---|---|---|---|---|---|---|---|---|---|---|
| | | $S_m \uparrow$ | $E_\xi^m \uparrow$ | $F_\beta^\omega \uparrow$ | $M \downarrow$ | $S_m \uparrow$ | $E_\xi^m \uparrow$ | $F_\beta^\omega \uparrow$ | $M \downarrow$ | $S_m \uparrow$ | $E_\xi^m \uparrow$ | $F_\beta^\omega \uparrow$ | $M \downarrow$ |
| 1 | baseline | .902 | .923 | .849 | .051 | .828 | .855 | .739 | .089 | .825 | .845 | .692 | .073 |
| 2 | +FR (w/o Skip) | .922 | .946 | .906 | .036 | **.857** | **.896** | .806 | **.066** | .879 | .911 | .812 | .042 |
| 3 | +FR (w/ Skip) | **.924** | **.948** | **.908** | **.034** | **.857** | .892 | **.807** | .067 | **.880** | **.914** | **.820** | **.040** |

The best results are highlighted in bold.

**Table 4.** Ablation and Sensitivity Analysis on Kernel/Dilation(K/D) Sizes of FR Module (on ECSSD).

| No | FR K/D Combination | ECSSD | | | | No | FR K/D Combination | ECSSD | | | |
|---|---|---|---|---|---|---|---|---|---|---|---|
| | | $S_m \uparrow$ | $E_\xi^m \uparrow$ | $F_\beta^\omega \uparrow$ | $M \downarrow$ | | | $S_m \uparrow$ | $E_\xi^m \uparrow$ | $F_\beta^\omega \uparrow$ | $M \downarrow$ |
| N0 | {1,3,5,7} | **.929** | **.955** | **.919** | **.031** | N0 | {1,3,5,7} | **.929** | 955 | **.919** | **.031** |
| N1 | {1,3,5} | .924 | .950 | .914 | .033 | N5 | {3,3,5,7} | **.929** | .953 | .918 | **.031** |
| N2 | {1,3,7} | .927 | .953 | .918 | .032 | N6 | {1,5,5,7} | .927 | .953 | .916 | .032 |
| N3 | {3,5,7} | .928 | .954 | .917 | .032 | N7 | {1,3,7,7} | .928 | .954 | .917 | .032 |
| N4 | {1,3,5,7,9} | .925 | .949 | .911 | .034 | N8 | {1,3,5,9} | **.929** | **.955** | .917 | **.031** |

The best results are highlighted in bold.

the lack of core features of medium-scale targets, and reducing branches fails to achieve effective coverage of full-scale targets. The significant performance degradation of N4 due to the introduction of scale 9 is attributed to the fact that an excessively large scale introduces redundant background information, leading to over-smoothing of features and thus a reduction in detection accuracy. To further verify the model's robustness to K/D parameters, we fixed the total number of scaled branches in the FR module to four and only replaced the K/D value of a single branch in the N0 baseline with its adjacent scale (N5~N8), completing the sensitivity analysis by observing the performance fluctuations caused by minor parameter adjustments. The results demonstrate that the performance metrics of N5~N8 exhibit only slight fluctuations compared with those of N0 (with the gap of most metrics within 0.002) without significant degradation, which proves that the proposed model has good robustness to minor variations in K/D parameters and validates the rationality and stability of the parameter settings.

## Comparison with advanced methods

For validation of the total performance of our model, we compared it with 14 advanced deep SOD models, including BASNet [27], EGNet [41], F3Net [42], ITSD [43], MINet [44], GCPANet [6], DFI [45], CII [46], ICON [14], MGuidNet [47], DIPONet [48], DSLRDNet [49], VST [50], and GLSTR [51]. For equitable comparison, all saliency maps for competing methods were sourced directly from the respective authors. We employed the same evaluation code to quantitatively assess these maps, calculating the metrics $S_m$, $E_\xi^m$, $F_\beta^\omega$, MAE, and FNR, and subsequently plotting the corresponding precision-recall and F-measure curves.

**Quantitative comparison.** Table 5 shows the quantitative evaluation results of our model and 14 advanced algorithms on five traditional baseline datasets in terms of $S_m$, $E_\xi^m$, $F_\beta^\omega$, and MAE (represented by M) metrics. The data reveals that our model consistently outperforms others. Out of a total of 20 metrics tested on five datasets, our model ranks first with 14 metrics, second with 3 metrics, and third with 2 metrics, which fully demonstrates the robustness of our model. It is worth mentioning that all four metrics of our model are optimal on HKU-IS. Meanwhile, all metrics of our model rank in the top three on ECSSD, PASCAL-S, and DUTS, indicating the superiority of our model. On OMRON, our model gets the first place in $S_m$, $E_\xi^m$, and $F_\beta^\omega$ metrics. The M metric is marginally less competitive, potentially attributable to the lower resolution of certain images within the OMRON dataset, which can impede precise feature extraction on those samples.

**Table 5. Experimental results of different models on five datasets.**

| Method | ECSSD | | | | PASCAL-S | | | | DUTS | | | | HKU-IS | | | | OMRON | | | |
|---|---|---|---|---|---|---|---|---|---|---|---|---|---|---|---|---|---|---|---|---|
| | $S_m \uparrow$ | $E_\xi^m \uparrow$ | $F_\beta^\omega \uparrow$ | $M \downarrow$ | $S_m \uparrow$ | $E_\xi^m \uparrow$ | $F_\beta^\omega \uparrow$ | $M \downarrow$ | $S_m \uparrow$ | $E_\xi^m \uparrow$ | $F_\beta^\omega \uparrow$ | $M \downarrow$ | $S_m \uparrow$ | $E_\xi^m \uparrow$ | $F_\beta^\omega \uparrow$ | $M \downarrow$ | $S_m \uparrow$ | $E_\xi^m \uparrow$ | $F_\beta^\omega \uparrow$ | $M \downarrow$ |
| **ResNet50-Based Methods** | | | | | | | | | | | | | | | | | | | | |
| BASNet | .916 | .943 | .904 | .037 | .838 | .879 | .793 | .076 | .866 | .895 | .803 | .040 | .909 | .943 | .889 | .032 | .836 | .865 | .751 | .056 |
| EGNet | .925 | .943 | .903 | .037 | .852 | .881 | .795 | .074 | .887 | .907 | .815 | .039 | .918 | .944 | .887 | .031 | .841 | .857 | .738 | **.053** |
| F3Net | .924 | .948 | .912 | .033 | .861 | .898 | .816 | **.061** | .888 | .920 | .835 | **.035** | .917 | .952 | .900 | .028 | .838 | .864 | .747 | **.053** |
| ITSD | .925 | .947 | .910 | .034 | .859 | .894 | .812 | .066 | .885 | .913 | .823 | .041 | .917 | .947 | .894 | .031 | .840 | .865 | .750 | .061 |
| MINet | .925 | .950 | .911 | .033 | .856 | .896 | .809 | .064 | .884 | .917 | .825 | .037 | .919 | .952 | .897 | .029 | .833 | .860 | .738 | .056 |
| GCPANet | .927 | .948 | .903 | .035 | .864 | .895 | .808 | .062 | .891 | .911 | .821 | .038 | .920 | .944 | .889 | .031 | .839 | .853 | .734 | .056 |
| DFI | .927 | .947 | .906 | .035 | .865 | .898 | .814 | .065 | .886 | .912 | .817 | .039 | .919 | .948 | .890 | .031 | .840 | .862 | .738 | .055 |
| CII | .926 | .948 | .913 | .033 | .865 | .901 | .821 | .062 | .887 | .917 | .830 | .037 | .920 | .952 | .902 | .029 | .839 | .865 | .749 | .054 |
| ICON-R | .929 | .954 | .918 | .032 | .861 | .899 | .818 | .064 | .888 | .924 | .836 | .037 | .920 | .953 | .902 | .029 | .844 | .876 | .761 | .057 |
| MGuidNet | .927 | .943 | .900 | .036 | **.869** | .897 | .812 | **.061** | .888 | .908 | .817 | .037 | .922 | .944 | .890 | .031 | .836 | .865 | .751 | .056 |
| DIPONet | .927 | .947 | .913 | .033 | .861 | .894 | .812 | .063 | .885 | .912 | .828 | .036 | .920 | .951 | .901 | .028 | .827 | .846 | .726 | **.053** |
| DSLRDNet | **.930** | .951 | .914 | .032 | - | - | - | - | **.895** | .923 | .834 | .036 | .922 | .951 | .896 | .029 | **.847** | .874 | .753 | **.053** |
| Ours-R | .929 | **.955** | **.919** | **.031** | .865 | **.902** | **.824** | .063 | .891 | **.925** | **.841** | .037 | **.923** | **.956** | **.908** | **.027** | **.847** | **.877** | **.768** | .056 |
| **Transformer-Based Methods** | | | | | | | | | | | | | | | | | | | | |
| VST | .932 | .951 | .910 | .033 | .872 | .902 | .816 | .061 | .896 | .919 | .828 | .037 | .928 | .952 | .897 | .029 | .850 | .871 | .755 | .058 |
| ICON-P | .940 | .964 | .933 | .024 | .882 | .921 | .847 | .051 | .917 | **.950** | .882 | **.022** | .935 | .967 | .925 | .022 | .865 | .896 | .793 | .047 |
| GLSTR | .942 | .961 | .930 | .025 | **.886** | .919 | .846 | .052 | **.919** | .944 | .872 | .027 | .936 | .961 | .914 | .024 | **.868** | .890 | .787 | **.046** |
| Ours-P | **.945** | **.967** | **.941** | **.023** | **.886** | **.924** | **.855** | **.049** | **.919** | .949 | **.887** | .026 | **.939** | **.968** | **.930** | **.021** | .867 | **.897** | **.802** | .050 |

The best results are highlighted in bold. '-' indicates missing data.

However, the model's overall performance remains demonstrably superior and robust. This performance advantage stems primarily from the TAMF module's efficacy in suppressing background interference via dynamic spatial and channel feature weight recalibration. Meanwhile, the FR module, utilizing four parallel convolutional branches and multi-scale dilated convolutions, significantly strengthens the model's ability to discern and represent features at varying scales.

Precision-Recall (PR) curves and F-measure curves serve to illustrate model performance across varying decision thresholds. Superior model performance is indicated when the PR curve approaches the ideal (1,1) point more closely and when the area under the F-measure curve is larger. Fig 6 shows PR curves, F-measure curves, and False Negative Rate (FNR) results for our method alongside advanced approaches across five benchmark datasets. Observably, our model's PR curve consistently converges closest to the (1,1) point on all datasets (Fig 6a). Simultaneously, the area integrated under its F-measure curve is markedly greater than that of competing models (Fig 6b). These results demonstrate that our model can not only maintain the simultaneous improvement of precision rate and recall rate under different thresholds but also maintain a better balance between precision rate and recall rate in the dynamic threshold adjustment process. As shown in Fig 6(c), our model has the lowest FNR value. Since 'FNR = 1 - Recall,' lower FNR values imply higher recall, affirming the advantage of our approach in SOD. Collectively, the results in Fig 6 demonstrate the proposed model's enhanced capability in efficiently detecting true salient objects while minimizing missed detections and false alarms, highlighting its robust overall efficacy.

**Visual comparison.** Fig 7 shows visual comparisons of the aforementioned models in different challenge scenarios, specifically for small objects (rows 1 and 2), large objects (rows 3 and 4), fine structures (rows 5 and 6), low contrast (rows 7 and 8), and multiple objects (rows 9 and 10). The results indicate that our model excels in detecting salient objects across diverse scenes, with higher precision than other methods. Taking large objects (rows 3 and 4) as an example: the image depicts a chair against a backdrop of a tree-lined path, and our model demonstrates accurate detection performance in this scenario. Although other competing models can also detect the chair, they fail to fully identify the entire chair; in contrast, our model can detect the chair completely. In scenarios featuring multiple objects, our approach exhibits enhanced capability in distinguishing foreground objects from the background. As evidenced in rows 9 and 10, our method successfully identifies complete salient objects, whereas competing models may omit details like the bird's beak or erroneously detect reflection artifacts such as branches. These visualizations confirm our method's capacity for precise SOD, generating detection maps with well-defined boundaries across diverse settings while effectively mitigating background noise interference.

## Failure cases

Although the model in this article has high performance, there are also cases of detection errors. As shown in Fig 8, in the 1st row of cases, the model confuses pillows, beds, and walls as salient objects; in the 2nd row, it incorrectly highlights two columns in the background while ignoring paintings, TVs, etc., on the wall; and in the 3rd row, it fails to identify three lamps as salient objects. Similarly, other competing models encounter challenges in these samples. We analyze that the possible reasons for failure may include insufficient differentiation between foreground and background regions, insufficient training samples (as in the 2nd and 3rd rows), and uncertainty in the annotation (as in the 1st row). To address these issues, future research could consider incorporating scene understanding and semantic information to enhance the model's recognition of semantic content.

## Conclusion

To address the two core problems of background noise interference and scale variation in salient object detection (SOD), we introduce complementary modules: the Triple Attention-guided Multi-resolution Fusion (TAMF) module and the Feature Refinement (FR) module. On the one hand, the TAMF module effectively suppresses the background noise

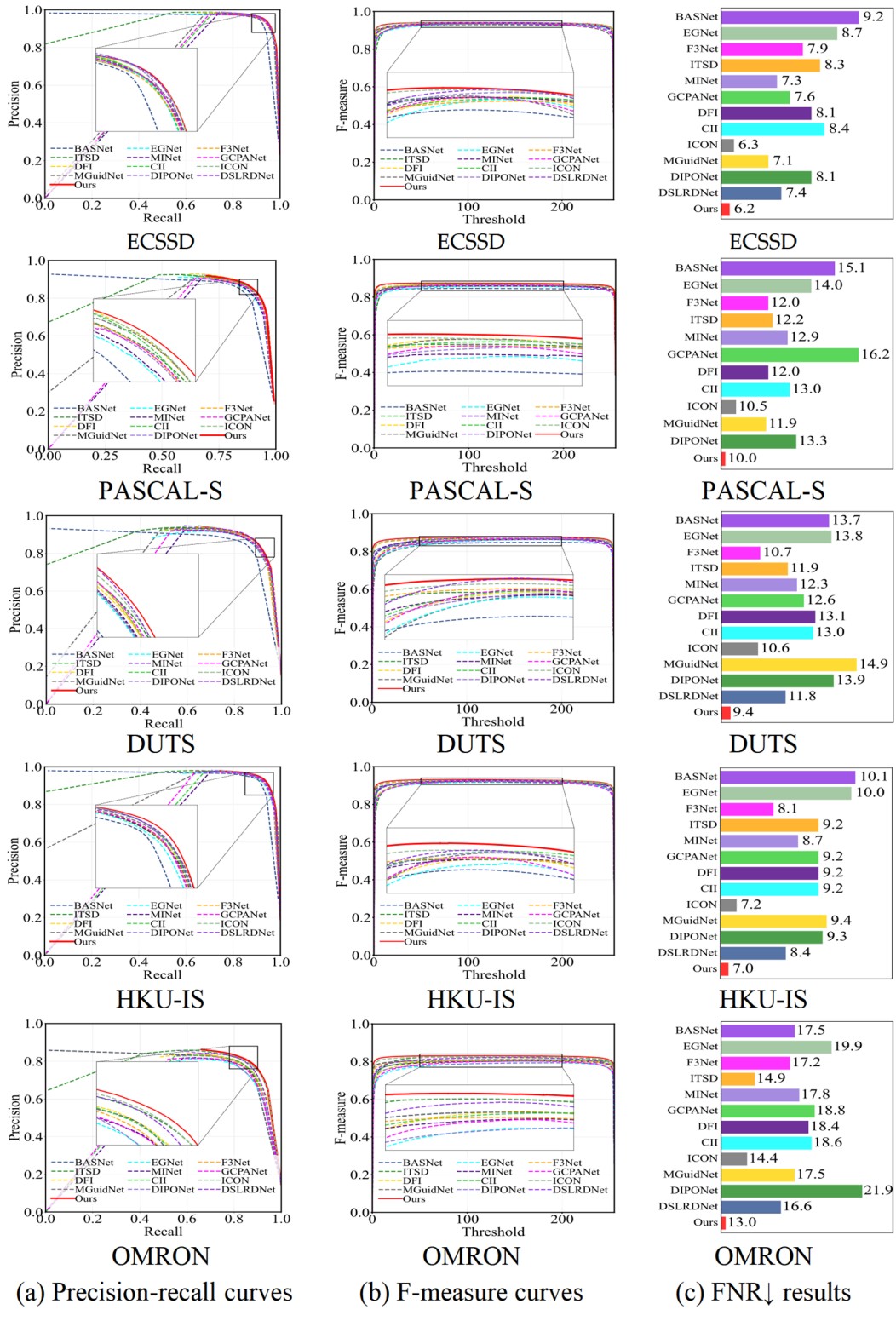

(a) Precision-recall curves (b) F-measure curves (c) FNR↓ results

**Fig 6**. Precision-recall, F-measure curves, and FNR↓ results.

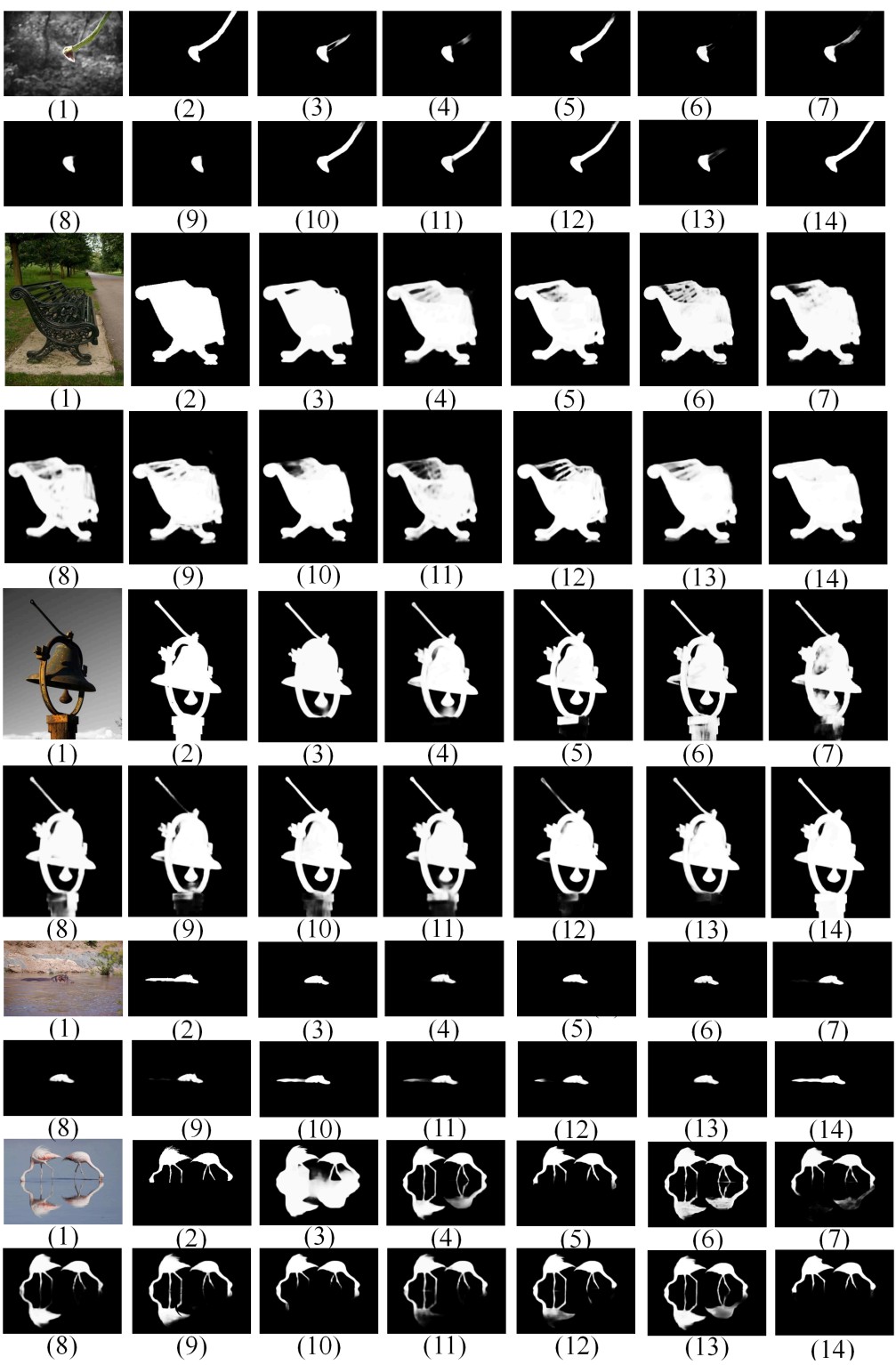

**Fig 7. Visual comparison of our method with advanced methods.** (1) Input image; (2) Ground truth; (3) BASNet; (4) EGNet; (5) F3Net; (6) ITSD; (7) MINet; (8) GCPANet; (9) DFI; (10) ICON; (11) MGuidNet; (12) DIPONet; (13) DSLRDNet; (14) Ours. *(To ensure the fairness of comparison, all visualization results are regenerated by running the official code of each method under a unified experimental setup. The use of input images complies with the licensing terms of the DUTS dataset (which is licensed under the 3-Clause BSD Open Source License, compatible with the CC BY 4.0 license; official download link: http://saliencydetection.net/duts/; source code repository: https://github.com/scott89/WSS, [30]).)*

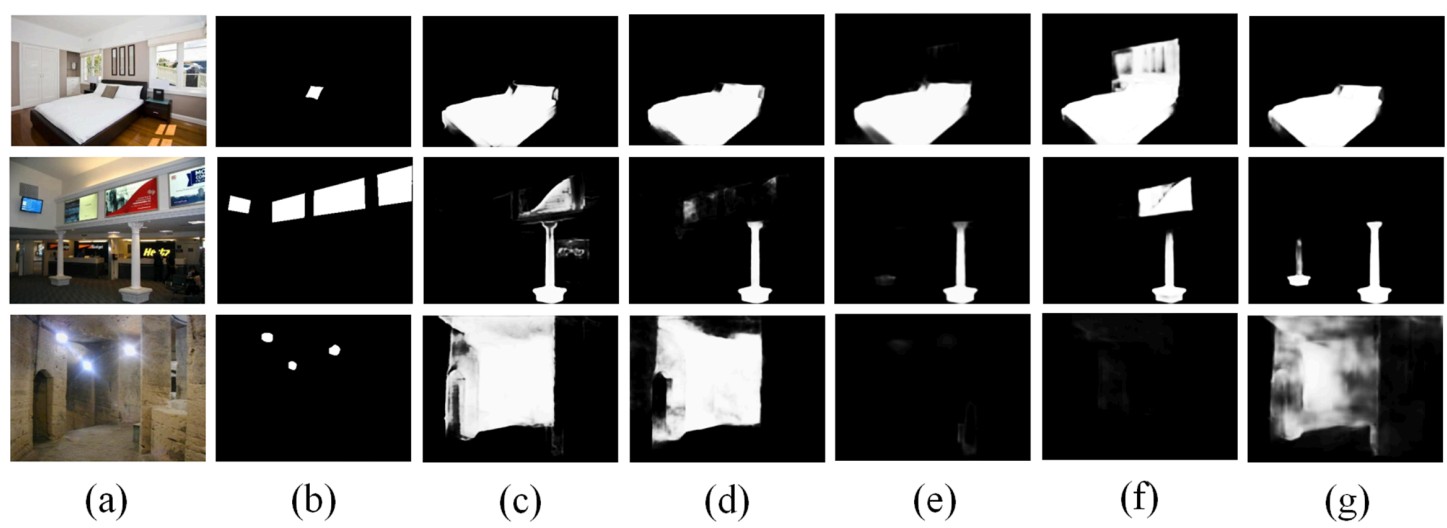

**Fig 8. Illustration of failure cases.** (a) Input image; (b) Ground truth; (c) ITSD; (d) MINet; (e) GCPANet; (f) ICON; (g) Ours.

interference by dynamically adjusting the spatial and channel weights, while concurrently improving the model's adaptability to multi-scale objects through multi-resolution fusion. On the other hand, the FR module combines four parallel convolutional branches and dilated convolutions at different scales, as well as the proposed triple attention mechanism, to achieve accurate capture and highlighting of salient object features, thereby successfully tackling the difficulties caused by scale variations. Through experimental validation on several mainstream benchmark datasets, our method demonstrates significant advantages over existing advanced methods.

## Author contributions

**Conceptualization:** Geng Wei, Mi Zhou.

**Data curation:** Geng Wei, Mi Zhou.

**Formal analysis:** Geng Wei, Mi Zhou, Jian Sun, Xiao Shi, Ming Yin.

**Funding acquisition:** Geng Wei.

**Investigation:** Geng Wei, Mi Zhou, Xinran Zhao, Xueyao Lin.

**Methodology:** Geng Wei, Mi Zhou.

**Project administration:** Geng Wei.

**Resources:** Geng Wei, Mi Zhou.

**Software:** Geng Wei, Mi Zhou.

**Supervision:** Geng Wei.

**Validation:** Geng Wei, Mi Zhou.

**Visualization:** Geng Wei, Mi Zhou.

**Writing – original draft:** Geng Wei, Mi Zhou.

**Writing – review & editing:** Geng Wei, Mi Zhou, Jian Sun, Xiao Shi, Ming Yin, Xinran Zhao, Xueyao Lin.

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
