## [Decision Letter · Decision Letter 0]

11 Dec 2025

PONE-D-25-52038Robust Salient Object Detection Based on Triple Attention-guided Multi-resolution Fusion and Feature RefinementPLOS One

Dear Dr. Wei,

Thank you for submitting your manuscript to PLOS ONE. After careful consideration, we feel that it has merit but does not fully meet PLOS ONE’s publication criteria as it currently stands. Therefore, we invite you to submit a revised version of the manuscript that addresses the points raised during the review process.

We look forward to receiving your revised manuscript.

Kind regards,

Wencheng Zhu, Ph.D

Academic Editor

PLOS One

[This research was funded by the Natural Science Foundation of Guangxi Province (Grants No. 2020GXNSFAA297184), the National Natural Science Foundation of China (Grant. No. 62161031).].

Additional Editor Comments:

This mansucript introduces a salient object detection approach in which the triple attention-guided multi-resolution fusion module and a feature refinement module are developed. Experiments on benchmarks show good performance. However, the authors should futher claim figures, references, ablation stuides and experimental details.

Reviewers' comments:

Reviewer's Responses to Questions

**Comments to the Author**

1. Is the manuscript technically sound, and do the data support the conclusions?

Reviewer #1: Yes

Reviewer #2: Yes

Reviewer #3: Yes

2. Has the statistical analysis been performed appropriately and rigorously?

Reviewer #1: Yes

Reviewer #2: Yes

Reviewer #3: Yes

3. Have the authors made all data underlying the findings in their manuscript fully available?

Reviewer #1: Yes

Reviewer #2: Yes

Reviewer #3: Yes

4. Is the manuscript presented in an intelligible fashion and written in standard English?

Reviewer #1: Yes

Reviewer #2: Yes

Reviewer #3: Yes

5. Review Comments to the Author

Reviewer #1: The manuscript is well-structured with clear logic and well-defined motivation. The solution proposed by the authors to address the critical challenges in the SOD field is theoretically sound. The experimental section is detailed, including ablation studies, quantitative comparisons, and qualitative visualizations. However, to meet the high standards required for publication, there is still room for improvement regarding the articulation of the method's novelty, the description of technical details, and the comprehensiveness of the experiments.

Experiments:

(1)The manuscript employs ResNet-50 as the backbone network. Although ResNet-50 serves as a classic and robust baseline, by 2025, Transformer-based backbones have become widely prevalent in the field of Salient Object Detection (SOD) and typically demonstrate superior capability in extracting global context information (e.g., [R1]). Therefore, the authors need to clarify the rationale behind selecting ResNet-50 as the backbone. Furthermore, to demonstrate that the effectiveness of the proposed TAMF and FR modules is not solely dependent on the CNN architecture, it is strongly recommended to include comparative experiments using a Transformer-based backbone. If the proposed modules can also yield performance improvements on a Transformer backbone, it would significantly strengthen the persuasiveness and state-of-the-art relevance of the paper.

(2)The Triple Attention (TA) module proposed in this paper, which integrates spatial, channel, and global attention, serves as a core innovation of this study. The "Related Works" section also notes that CBAM similarly utilizes channel and spatial attention. Given that CBAM is a classic dual-attention mechanism, it is natural for readers to question the specific extent of improvement brought by the additional "global attention" branch and the proposed fusion strategy compared to the standard CBAM. Therefore, it is recommended to include a comparative visualization experiment within the ablation study: replacing the TA module with the standard CBAM module while keeping all other settings unchanged. If the visual results demonstrate that TA outperforms CBAM, this would directly and compellingly attest to the necessity and effectiveness of introducing the "Global branch", thereby significantly enhancing the persuasiveness of the module's design.

Figures

(3)Figure 1 illustrates the model framework; however, the connecting lines between various components intersect, resulting in a somewhat visually cluttered presentation. It is therefore recommended to optimize the routing of the lines in Figure 1 to improve clarity.

(4)Figure 6 includes PR curves, F-measure curves, and FNR bar charts, but the arrangement appears overly crowded. It is recommended to adjust the layout of the legends and split the content into three separate figures.

Equations:

(5)The subscript notation 'Conv(2n−1)×(2n−1)' in Equation (9) appears slightly cumbersome. It is recommended to define the kernel size kn=2n−1 prior to the equation and then use Convkn×kn within the formula, which would be visually more concise.

[R1] Jiang, Z., Yu, L., Han, Y., Li, J., & Niu, F. (2025). Global-aware Interaction Network for RGB-D salient object detection. Neurocomputing, 621, 129204.

Reviewer #2: The authors propose an attention-based method for salient object detection. A Triple Attention-guided Multi-resolution Fusion (TAMF) module and a Feature Refinement (FR) module are devised. Comprehensive evaluations and ablation studies on five widely-used SOD benchmarks are given. However, the authors emphasize that addressing the following issues will significantly enhance the quality of this paper.

1. In related works, the author may consider categorizing salient object detection as follows: RGB, RGB-D, and RGB-T salient object detection for natural images; as well as salient object detection for optical remote sensing imagery. Some of the following papers may be referenced:

[1]CATNet: A Cascaded and Aggregated Transformer Network for RGB-D Salient Object Detection. IEEE Trans. Multim. 26: 2249-2262 (2024)

[2]Cross-Modal Fusion and Progressive Decoding Network for RGB-D Salient Object Detection. Int. J. Comput. Vis. 132(8): 3067-3085 (2024)

[3]MAGNet: Multi-scale Awareness and Global fusion Network for RGB-D salient object detection. Knowl. Based Syst. 299: 112126 (2024)

[4]LESOD: Lightweight and efficient network for RGB-D salient object detection. Pattern Recognit. 171: 112103 (2026)

[5]Highly Efficient RGB-D Salient Object Detection With Adaptive Fusion and Attention Regulation. IEEE Trans. Circuits Syst. Video Technol. 35(4): 3104-3118 (2025)

[6]ORSIDiff: Diffusion Model for Salient Object Detection in Optical Remote Sensing Images. IEEE Trans. Geosci. Remote. Sens. 63: 1-15 (2025)

[7]Exploring a Lightweight and Efficient Network for Salient Object Detection in ORSI. IEEE Trans. Geosci. Remote. Sens. 63: 1-14 (2025)

2. Figures require significant improvement in quality.

Reviewer #3: 1. The ablation study in Table 1 is too coarse. It only validates the inclusion of the whole TAMF or FR modules (Baseline vs. +TAMF vs. +FR). There is no internal ablation to justify the specific design choices. For instance: In TAMF, does the "Global Attention" branch actually contribute to performance compared to a standard CBAM? In FR, does the "inter-branch hopping mechanism" provide a statistically significant improvement over standard parallel dilated convolutions? Without these fine-grained analyses, it is impossible to determine which components are actually effective.

2. The authors state they employ the global feature vector $S_g$ as "a form of self-attention". However, Eq. 5 ($S_{gap} = S_g \odot S_4 + S_4$) describes a simple channel-wise re-weighting (broadcasting a vector to a tensor). Describing this as "self-attention" is misleading in the context of modern literature where self-attention typically refers to spatial non-local interactions (e.g., $Q, K, V$ matrices). In Equation 9, the notation mixes $DConv$ and $Conv$ in a way that is difficult to parse.

3. In Fig 1, the legend incorrectly spells "Upsample" as "Upsampe".

6. PLOS authors have the option to publish the peer review history of their article (what does this mean?). If published, this will include your full peer review and any attached files.

Reviewer #1: No

Reviewer #2: No

Reviewer #3: No

You may also use PLOS’s free figure tool, NAAS, to help you prepare publication quality figures: https://journals.plos.org/plosone/s/figures#loc-tools-for-figure-preparation

---

## [Author Response · Author response to Decision Letter 1]

3 Jan 2026

Subject: Response to Reviewer Comments – Manuscript ID [PONE-D-25-52038]

Dear Reviewer #1,

Thank you for your insightful and constructive feedback on our manuscript. We sincerely appreciate the time and effort you have dedicated to reviewing our work. Your comments have been invaluable in helping us improve the quality and clarity of the paper. Below, we provide a point-by-point response to each of your concerns and detail the corresponding revisions made to the manuscript.

Reviewer Comment 1:

The manuscript employs ResNet-50 as the backbone network. Although ResNet-50 serves as a classic and robust baseline, by 2025, Transformer-based backbones have become widely prevalent in the field of Salient Object Detection (SOD) and typically demonstrate superior capability in extracting global context information (e.g., [R1]). Therefore, the authors need to clarify the rationale behind selecting ResNet-50 as the backbone. Furthermore, to demonstrate that the effectiveness of the proposed TAMF and FR modules is not solely dependent on the CNN architecture, it is strongly recommended to include comparative experiments using a Transformer-based backbone. If the proposed modules can also yield performance improvements on a Transformer backbone, it would significantly strengthen the persuasiveness and state-of-the-art relevance of the paper.

Author Response 1:

We sincerely appreciate the reviewers’ valuable suggestion. We fully agree that evaluating our proposed modules on a Transformer backbone is crucial to verifying their generalizability beyond CNN architectures. To this end, we conducted additional control experiments using PVTv2 as a representative Transformer backbone, with detailed quantitative results presented in Table 4 (Section 4.4) of the revised manuscript. The experimental results validate two core values: our proposed TAMF and FR modules consistently yield performance gains when integrated into the PVTv2 backbone, confirming their architecture-agnostic effectiveness that is not limited to CNN backbones such as ResNet-50; additionally, we supplemented comparisons with recent Transformer-based SOD methods (including VST, ICON, and GLSTR). As shown in Table 4, our full model (PVTv2 + TAMF + FR) achieves superior or highly competitive results across five standard benchmark datasets, fully demonstrating its advanced performance and practical value in the field.

Reviewer Comment 2:

The Triple Attention (TA) module proposed in this paper, which integrates spatial, channel, and global attention, serves as a core innovation of this study. The "Related Works" section also notes that CBAM similarly utilizes channel and spatial attention. Given that CBAM is a classic dual-attention mechanism, it is natural for readers to question the specific extent of improvement brought by the additional "global attention" branch and the proposed fusion strategy compared to the standard CBAM. Therefore, it is recommended to include a comparative visualization experiment within the ablation study: replacing the TA module with the standard CBAM module while keeping all other settings unchanged. If the visual results demonstrate that TA outperforms CBAM, this would directly and compellingly attest to the necessity and effectiveness of introducing the "Global branch", thereby significantly enhancing the persuasiveness of the module's design.

Author Response 2:

We sincerely appreciate the reviewers’ critical question regarding the necessity of the global attention branch in our Triple Attention (TA) module. To address this concern, we conducted the suggested ablation experiment and added a new Table 2 in Section 4.3 (Ablation Study). To verify the superiority of the TA sub-module over CBAM while eliminating interference from the FR module (avoiding ambiguity about the source of performance gains), we excluded FR from the experiment and only replaced TA with the standard CBAM. Quantitative results show that our TA module outperforms CBAM on most metrics across the ECSSD, PASCAL-S, and DUTS datasets. This performance advantage directly confirms the effectiveness of the introduced global attention branch and our novel fusion strategy — unlike CBAM’s dual-attention mechanism, the global branch enables the model to capture long-range dependencies and global scene context, thereby more effectively suppressing background noise and enhancing salient object consistency, especially in complex scenes. We have provided detailed analysis of these findings in the revised text accompanying new Table 2.

Reviewer Comment 3:

Figure 1 illustrates the model framework; however, the connecting lines between various components intersect, resulting in a somewhat visually cluttered presentation. It is therefore recommended to optimize the routing of the lines in Figure 1 to improve clarity.

Author Response 3 :

We sincerely appreciate the reviewers’ constructive feedback on the architecture diagram. We agree that diagram clarity is critical for reader understanding, and have simplified the connection lines in Figure 1 as suggested, significantly reducing crossings and overlaps. The revised Figure 1 presents our proposed framework more clearly and intuitively, facilitating readers’ comprehension of the model. The updated Figure 1 is included in the revised manuscript.

Reviewer Comment 4:

Figure 6 includes PR curves, F-measure curves, and FNR bar charts, but the arrangement appears overly crowded. It is recommended to adjust the layout of the legends and split the content into three separate figures.

Author Response 4:

We appreciate the reviewers’ suggestion regarding Figure 6. After careful consideration, we retained the combined subplot format while alleviating the overcrowding issue through optimizations. The primary purpose of this figure is to enable readers to directly visually compare performance across three key metrics on the same dataset. Presenting them in a single row allows readers to immediately correlate performance trends of all metrics without flipping between separate pages or figures, enhancing analysis efficiency. To address the noted overcrowding, we increased the horizontal and vertical spacing between subplots. We believe these adjustments not only effectively mitigated visual overcrowding but also preserved the figure’s core value for comprehensive performance evaluation, and we hope the revised version meets your approval.

Reviewer Comment 5:

The subscript notation 'Conv(2n−1)×(2n−1)' in Equation (9) appears slightly cumbersome. It is recommended to define the kernel size kn=2n−1 prior to the equation and then use Convkn×kn within the formula, which would be visually more concise.

Author Response 5:

Thank you for the suggestion to optimize the mathematical notation. We have adopted it. In the revised manuscript, we have added the definition kn=2n−1 before Equation (9) and updated the equation to the more concise form Convkn×kn. This change enhances the readability and conciseness of the formula.

Dear Reviewer #2,

Thank you for your positive evaluation of our work and for providing such constructive and specific feedback. We greatly appreciate the time you have taken to review our manuscript. Your suggestions have been instrumental in enhancing the quality and clarity of our paper. Below, we address each of your comments in detail.

Reviewer Comment 1:

In related works, the author may consider categorizing salient object detection as follows: RGB, RGB-D, and RGB-T salient object detection for natural images; as well as salient object detection for optical remote sensing imagery.

Author Response 1:

We appreciate the reviewers’ valuable and constructive suggestions on the organization of the "Related Works" section. We agree that adopting a clearer modality-oriented classification will significantly enhance the structure and comprehensiveness of the literature review. To this end, we have revised Section 2.1 in the revised manuscript, strictly adopting the proposed logical classification: Salient Object Detection (SOD) in natural images (subdivided into RGB-based SOD, RGB-D SOD, and RGB-T SOD) and SOD in optical remote sensing images (ORSI SOD). Additionally, we have carefully reviewed and incorporated all seven key references provided ([1]-[7]), and these recent relevant studies have been integrated into Section 2.1 to present a more cutting-edge and comprehensive overview of the state-of-the-art.

Reviewer Comment 2:

Figures require significant improvement in quality.

Author Response 2:

We sincerely thank the reviewers for pointing out the issues with figure quality. We have comprehensively upgraded all figures in the manuscript to meet publication standards: All diagrams have been recreated or exported as vector graphics, ensuring clarity and losslessness at any scale or resolution; all qualitative result figures (e.g., saliency map visualizations) have been regenerated using the highest quality settings and saved in lossless formats.

Dear Reviewer #3,

Thank you for your thorough review and for raising these critical points, which have significantly helped us refine both the technical justification and presentation of our work. We have carefully addressed each of your comments as detailed below.

Reviewer Comment 1:

The ablation study in Table 1 is too coarse. It only validates the inclusion of the whole TAMF or FR modules (Baseline vs. +TAMF vs. +FR). There is no internal ablation to justify the specific design choices. For instance: In TAMF, does the "Global Attention" branch actually contribute to performance compared to a standard CBAM? In FR, does the "inter-branch hopping mechanism" provide a statistically significant improvement over standard parallel dilated convolutions? Without these fine-grained analyses, it is impossible to determine which components are actually effective.

Author Response 1:

We sincerely appreciate the reviewers’ core comment. We agree that detailed analysis is required to verify the necessity of each key component, and have added two comprehensive ablation experiments in the revised manuscript. To evaluate the contribution of the global attention branch in our Triple Attention (TA) module, we replaced TA with the standard CBAM while keeping all other components unchanged. Quantitative results in new Table 2 demonstrate that our TA module outperforms CBAM across all metrics, confirming the effectiveness of the integrated global attention mechanism and our specific fusion strategy. To assess the importance of the inter-branch skip mechanism in the Feature Refinement (FR) module, we compared the full FR module with a variant where skip connections are removed (retaining only parallel dilated convolution branches). As shown in new Table 3, quantitative results verify that the skip mechanism yields statistically significant performance gains, validating its role in facilitating complementary information flow and enhancing feature refinement. These new tables (Tables 2 and 3) directly address your concern by providing detailed evidence to justify the core design choices, and we have updated Section 4.3 to discuss these findings.

Reviewer Comment 2:

The authors state they employ the global feature vector $S_g$ as "a form of self-attention". However, Eq. 5 ($S_{gap} = S_g \odot S_4 + S_4$) describes a simple channel-wise re-weighting (broadcasting a vector to a tensor). Describing this as "self-attention" is misleading in the context of modern literature where self-attention typically refers to spatial non-local interactions (e.g., $Q, K, V$ matrices). In Equation 9, the notation mixes $DConv$ and $Conv$ in a way that is difficult to parse.

Author Response 2:

We appreciate your precise and helpful feedback on terminology and equations. Regarding the clarification of "self-attention": You are absolutely correct. Using this term in Equation 5 is inaccurate in the contemporary context. We have revised the relevant text (Section 3.2) to provide a more precise description, which now reads: "Instead, we use Sg as a gate for channel-wise reweighting to recalibrate and refine features via residual connection." We appreciate this suggestion, as it enhances the technical accuracy of our manuscript. Regarding the clarification of Equation 9: Thank you for pointing out the potential ambiguity. We have revised Equation 9 to improve clarity.

Reviewer Comment 3:

In Fig 1, the legend incorrectly spells "Upsample" as "Upsampe".

Author Response 3:

We appreciate the reviewers’ meticulous attention to detail. This typo has been corrected in the revised Figure 1, which is now correctly labeled as "Upsample".

Sincerely,

Geng Wei

Corresponding Author: wei_geng@nnnu.edu.cn

Affiliation: School of Physics and Electronics, Nanning Normal University, China

---

## [Decision Letter · Decision Letter 1]

16 Jan 2026

PONE-D-25-52038R1Robust Salient Object Detection Based on Triple Attention-guided Multi-resolution Fusion and Feature RefinementPLOS One

Dear Dr. Wei,

Thank you for submitting your manuscript to PLOS ONE. After careful consideration, we feel that it has merit but does not fully meet PLOS ONE’s publication criteria as it currently stands. Therefore, we invite you to submit a revised version of the manuscript that addresses the points raised during the review process. The authors should address three aspects: First, the sensitivity analysis of kernel sizes. Second, ablation on the global branch. Finally, missing references on SOD.

We look forward to receiving your revised manuscript.

Kind regards,

Wencheng Zhu, Ph.D

Academic Editor

PLOS One

Journal Requirements:

Reviewers' comments:

Reviewer's Responses to Questions

**Comments to the Author**

1. If the authors have adequately addressed your comments raised in a previous round of review and you feel that this manuscript is now acceptable for publication, you may indicate that here to bypass the “Comments to the Author” section, enter your conflict of interest statement in the “Confidential to Editor” section, and submit your "Accept" recommendation.

Reviewer #1: All comments have been addressed

Reviewer #3: (No Response)

2. Is the manuscript technically sound, and do the data support the conclusions?

Reviewer #1: Yes

Reviewer #3: Yes

3. Has the statistical analysis been performed appropriately and rigorously?

Reviewer #1: Yes

Reviewer #3: Yes

4. Have the authors made all data underlying the findings in their manuscript fully available?

Reviewer #1: Yes

Reviewer #3: Yes

5. Is the manuscript presented in an intelligible fashion and written in standard English?

Reviewer #1: Yes

Reviewer #3: Yes

6. Review Comments to the Author

Reviewer #1: (No Response)

Reviewer #3: 1. The choice of kernel sizes $k=\{1, 3, 5, 7\}$ and dilation rates is presented without justification or sensitivity analysis. It is unclear if these values are optimal or how sensitive the model is to changes in these parameters.

2. Comparing "Baseline + TA" vs. "Baseline + CBAM" (Table 2) proves that the entire TA module works better than CBAM. However, it does not prove why. It does not isolate whether the gain comes from the "Global Attention" branch specifically or simply from the parallel arrangement of the branches. To rigorously justify the "Triple" attention claim, an ablation removing only the Global Attention branch from the TA module (i.e., "Baseline + Spatial + Channel") is required. This would verify if the "Global" branch adds unique value beyond standard Channel+Spatial attention.

3. Some works about attention and SOD should be cited in this paper to make this submission more comprehensive, such as

10.1109/TPAMI.2024.3511621, 10.1016/j.patcog.2022.108792, 10.1145/3394171.3413884,10.1145/3581783.3612221,

10.24963/ijcai.2025/693.

7. PLOS authors have the option to publish the peer review history of their article (what does this mean?). If published, this will include your full peer review and any attached files.

Reviewer #1: No

Reviewer #3: No

---

## [Author Response · Author response to Decision Letter 2]

28 Jan 2026

Subject: Response to Reviewer Comments – Manuscript ID [PONE-D-25-52038R1]

Dear Dr. Wencheng Zhu,

Additional Notes on Reference List Changes:

Thank you for handling our manuscript and granting us the opportunity to revise it. We have conducted a comprehensive review of the reference list to ensure its completeness and accuracy, with the specific measures implemented as follows:

Verification of all references: We have cross-checked all citations in the manuscript against their original sources, confirming the accuracy of authors’ names, article titles, journal names, volumes, issues, page numbers, publication years and DOIs for each entry.

Screening for retracted papers: We have specifically screened the retraction status of all cited articles using the Retraction Watch database and official publisher websites, and confirm that no retracted papers are cited in the manuscript.

Added references: Following the reviewers’ suggestions and to strengthen the literature review, we have added 5 highly relevant, recently published references (e.g., [22], [23], [24], [25], [26]). All these newly added references have been verified to be unretracted. Changes are fully reflected in the revised reference list and corresponding in-text citations, and highlighted in the tracked-changes manuscript.

Format compliance: The reference list has been formatted in strict accordance with the PLOS ONE Citation Format Guidelines.

We confirm that the revised manuscript now fully complies with the reference policies of PLOS ONE. Please do not hesitate to inform us if any further adjustments are required.

Dear Reviewer #3,

Thank you for your positive evaluation of our work and for providing such constructive and specific feedback. We greatly appreciate the time you have taken to review our manuscript. Your suggestions have been instrumental in enhancing the quality and clarity of our paper. Below, we address each of your comments in detail.

Reviewer Comment 1:

The choice of kernel sizes $k=\{1, 3, 5, 7\}$ and dilation rates is presented without justification or sensitivity analysis. It is unclear if these values are optimal or how sensitive the model is to changes in these parameters.

Author Response 1:

To address the concern raised by the reviewer, we have supplemented targeted ablation experiments and summarized the results in Table 4, which fully responds to this comment.

To verify the optimality of the kernel size/dilation rate (K/D) parameter selection in the FR module and analyze the model’s sensitivity to variations in such parameters, we designed two groups of ablation experiments on the ECSSD dataset. The first group (N0–N4) includes the baseline scheme N0 (K/D = {1,3,5,7}) and its variants with pruned or extended scale branches. Experimental results show that all performance metrics of N1 ({1,3,5}), N2 ({1,3,7}) and N3 ({3,5,7}) are slightly lower than those of N0, which confirms that the fine-grained scale 1, medium scale 5 and large scale 7 are all critical scales for the model—removing any of these scales will lead to the loss of target features at the corresponding scale and thus fail to achieve effective coverage of full-scale targets. In contrast, N4 ({1,3,5,7,9}) with an additional scale of 9 exhibits a significant performance drop, because an excessively large scale introduces redundant background information, resulting in feature over-smoothing and a subsequent reduction in detection accuracy. This result fully demonstrates the rationality of selecting the four-branch K/D set {1,3,5,7} and clarifies the reasons for not adopting fewer/more branches or larger scales.

The second group (N0, N5–N8) focuses on the sensitivity analysis of minor parameter adjustments: we fixed the four-scale branch structure of the FR module and only replaced the K/D value of a single branch in N0 with its adjacent scale to obtain N5 ({3,3,5,7}), N6 ({1,5,5,7}), N7 ({1,3,7,7}) and N8 ({1,3,5,9}). The results indicate that all performance metrics of N5–N8 show only slight fluctuations compared with those of N0 (with the gap of most metrics within 0.002) without significant degradation. This proves that the proposed model has good robustness to minor variations in K/D parameters and further validates the stability and rationality of the original parameter settings.

All the above supplementary ablation experiments and detailed analyses have been added to the revised manuscript (corresponding to Table 4), which fully addresses the questions raised by Reviewer 3 regarding the optimality of K/D parameter selection and the model’s parameter sensitivity.

Reviewer Comment 2:

Comparing "Baseline + TA" vs. "Baseline + CBAM" (Table 2) proves that the entire TA module works better than CBAM. However, it does not prove why. It does not isolate whether the gain comes from the "Global Attention" branch specifically or simply from the parallel arrangement of the branches. To rigorously justify the "Triple" attention claim, an ablation removing only the Global Attention branch from the TA module (i.e., "Baseline + Spatial + Channel") is required. This would verify if the "Global" branch adds unique value beyond standard Channel+Spatial attention.

Author Response 2:

In response to your comment on verifying the unique value of the Global Attention branch in the TA module, we have supplemented the ablation experiment Baseline + TA (w/o Global) + MF in Table 2 for rigorous validation. The experimental results show that the performance of Baseline + TA + MF is the optimal among the relevant variants, and it is significantly superior to that of Baseline + TA (w/o Global) + MF. This direct comparison fully demonstrates that the Global Attention branch in the TA module provides unique and indispensable performance gains that cannot be achieved by the combination of only Spatial and Channel Attention branches. It thus validates the rationality of the "Triple" attention design of the TA module, and confirms that the Global Attention branch plays a crucial role in enhancing the model's ability to capture global contextual information of salient targets, further improving the overall detection performance of the model. All the supplementary experimental data has been updated in the revised Table 2 of the manuscript.

Reviewer Comment 3:

Some works about attention and SOD should be cited in this paper to make this submission more comprehensive, such as

10.1109/TPAMI.2024.3511621, 10.1016/j.patcog.2022.108792, 10.1145/3394171.3413884,10.1145/3581783.3612221,

10.24963/ijcai.2025/693.

Author Response 3:

We sincerely thank you for pointing out this important oversight in the literature review and for providing highly relevant references. We have incorporated all the five recommended references into the revised manuscript. Specifically, we have added a concise discussion in Section 2.2 to illustrate how these studies demonstrate the cutting-edge applications of attention mechanisms in various scenarios, and these additions have significantly enhanced the comprehensiveness of the research discussion in this paper.

Sincerely,

Geng Wei

Corresponding Author: wei_geng@nnnu.edu.cn

Affiliation: School of Physics and Electronics, Nanning Normal University, China

---

## [Editor Report · Decision Letter 2]

1 Feb 2026

Robust Salient Object Detection Based on Triple Attention-guided Multi-resolution Fusion and Feature Refinement

PONE-D-25-52038R2

Dear Dr. Geng,

We’re pleased to inform you that your manuscript has been judged scientifically suitable for publication and will be formally accepted for publication once it meets all outstanding technical requirements.

Kind regards,

Wencheng Zhu, Ph.D

Academic Editor

PLOS One
---

## [Editor Report · Acceptance letter]

PONE-D-25-52038R2

PLOS One

Dear Dr. Wei,

I'm pleased to inform you that your manuscript has been deemed suitable for publication in PLOS One. Congratulations! Your manuscript is now being handed over to our production team.

Kind regards,

on behalf of

Dr. Wencheng Zhu

Academic Editor

PLOS One